# Dysuricemia—A New Concept Encompassing Hyperuricemia and Hypouricemia

**DOI:** 10.3390/biomedicines11051255

**Published:** 2023-04-23

**Authors:** Naoyuki Otani, Motoshi Ouchi, Einosuke Mizuta, Asuka Morita, Tomoe Fujita, Naohiko Anzai, Ichiro Hisatome

**Affiliations:** 1Department of Cardiology, Dokkyo Medical University Nikkyo Medical Center, Nikko 321-1298, Tochigi, Japan; 2Department of Pharmacology and Toxicology, Dokkyo Medical University School of Medicine, Mibu 321-0293, Tochigi, Japan; 3Department of Cardiology, Sanin Rosai Hospital, Yonago 683-8605, Tottori, Japan; 4Department of Pharmacology, Chiba University Graduate School of Medicine, Chiba 260-8670, Chiba, Japan; 5Yonago Medical Center, National Hospital Organization, Yonago 683-0006, Tottori, Japan

**Keywords:** dysuricemia, hyperuricemia, hypouricemia, uric acid, J-shaped curve, antioxidant

## Abstract

The importance of uric acid, the final metabolite of purines excreted by the kidneys and intestines, was not previously recognized, except for its role in forming crystals in the joints and causing gout. However, recent evidence implies that uric acid is not a biologically inactive substance and may exert a wide range of effects, including antioxidant, neurostimulatory, proinflammatory, and innate immune activities. Notably, uric acid has two contradictory properties: antioxidant and oxidative ones. In this review, we present the concept of “dysuricemia”, a condition in which deviation from the appropriate range of uric acid in the living body results in disease. This concept encompasses both hyperuricemia and hypouricemia. This review draws comparisons between the biologically biphasic positive and negative effects of uric acid and discusses the impact of such effects on various diseases.

## 1. Introduction

Gout is one of the most well-known and ancient diseases, which continues to affect humans today. Recent data from the 2007–2016 National Health and Nutrition Examination Survey (NHANES) revealed that the prevalence of gout in the United States is 3.9%, which is equivalent to approximately 9.2 million people [1].

Uric acid was first identified in urine by Scheele in 1776 and later extracted from the tophi of gout by Wollastone in 1787 [2]. In 1848, Garrod discovered that patients with gout had elevated blood glucose levels [3]. Gout is a type of inflammatory arthritis that occurs when sodium uric acid crystallizes in the joints and is preceded by hyperuricemia. Historically, hyperuricemia was considered synonymous with gout. Uric acid was thought to be a form of waste excreted from the kidneys and intestinal tract. When serum uric acid rises above the dissolution limit, it crystallizes and precipitates in the joints, causing inflammation. If it precipitates in the urinary tract, it forms stones. Therefore, serum uric acid levels were measured only when gouty arthritis and uric acid urolithiasis were suspected. Traditionally, asymptomatic hyperuricemia was generally considered a benign disease that did not require treatment [4,5]. However, in recent years, it has become increasingly clear that uric acid is not a biologically inert substance but has various biological functions. Therefore, both high and low uric acid levels can affect an organism, even if the condition is asymptomatic.

Hyperuricemia is strongly associated with lifestyle-related diseases such as hypertension [6,7], type 2 diabetes mellitus [8], and metabolic syndrome [9]. It may contribute to myocardial infarction, an atherosclerotic disease [10,11]. In addition, gout caused by hyperuricemia is extremely painful and reduces the quality of life [12] and physical function [13]. Conversely, a study implied that hypouricemia causes exercise-induced renal failure and urinary stones [14]. Therefore, it is now understood that both elevated and reduced uric acid levels can have adverse effects on the living body. In this context, we propose the concept of “uricemia,” which encompasses both hyperuricemia and hypouricemia. This review outlines the conflicting biological roles of uric acid and discusses its impact on various disease states.

## 2. Definition and Epidemiology

Hyperuricemia is determined by the solubility of uric acid and is defined as a serum uric acid level greater than 7.0 mg/dL in both men and women [14]. In the NHANES, the prevalence rates of gout and hyperuricemia remained substantial in the U.S. between 2007 and 2016, with gout having a prevalence of 3.9% (9.2 million individuals) among U.S. adults from 2015 to 2016. The mean serum uric acid level was 6.0 mg/dL in men and 4.8 mg/dL in women, and the prevalence of hyperuricemia was 20.2% in men and 20.0% in women [1]. The data show that the prevalence of hyperuricemia has been stable over the last 10 years, but it still persists at a considerable rate.

Although there is no established definition of hypouricemia, a value of 2.0 mg/dL or less is generally used as a reference value for this condition [14]. Hypouricemia is less common than hyperuricemia, and in Japan, its prevalence was reported to be 0.2% in men and 0.4% in women [15]. Hypouricemia was also found to be a rare outcome in Germany, affecting 0.09% of the population (0% in men; 0.27% in women) [16]. Regardless of race or region, hypouricemia has been found to be more prevalent in women than in men.

### 2.1. Serum Uric Acid Regulation

Blood uric acid levels depend on the balance between the exogenous production of uric acid via dietary factors (including purine intake) and its endogenous production via purine metabolism, reabsorption and excretion by the kidney, and excretion by the intestine [17]. Thus, uric acid is homeostasis through the complicated process of production, secretion, and reabsorption in the kidney and excretion in the intestines. Uric acid is produced by the metabolism of endogenous purines (approximately 300–400 mg synthesized daily) and exogenous purines (approximately 300 mg from the diet), totaling 1200 mg in healthy men (600 mg in healthy women) on purine-free diets. The purine nucleobases adenine and guanine are components of both ribonucleic acid (RNA) and deoxyribonucleic acid (DNA). Therefore, the breakdown of RNA and DNA increases blood uric acid levels. Many hematological diseases, such as acute myelogenous leukemia, involve the production of excessive uric acid as tumor cells proliferate and collapse, causing hyperuricemia. Hyperuricemia is also caused by the rapid disintegration of tumor cells by antitumor agents. This is called tumor lysis syndrome and is complicated by hyperkalemia and hyperphosphatemia. Hyperuricemia associated with hematological diseases reflects tumor volume. It may also indicate a response to treatment with antitumor agents. Rapidly elevated uric acid levels can lead to renal failure, which is treated with uric acid inhibitors or uric acid-degrading enzymes, depending on the risk of developing the disease [18]. In addition, uric acid is produced by the breakdown of adenosine triphosphate (ATP) through the metabolism of fructose and alcohol. ATP is an important source of energy for intracellular reactions. In terms of dietary effects, animal protein contributes significantly to the intake of this purine. In addition, serum uric acid levels are increased by the intake of glutamic acid, which is metabolized to uric acid in the liver, and by the intake of foods high in purines and uric acid itself. Foods that affect serum uric acid levels include alcohol, meat and seafood, soft drinks, dairy products, coffee, and vitamin C. A purine-free diet reduces uric acid excretion via urine by approximately 40% [19], indicating that dietary purine contributes greatly to serum uric acid levels.

Adenosine products are converted to hypoxanthine, and hypoxanthine is oxidized to xanthine and then uric acid by xanthine oxidoreductase (XOR). Approximately two-thirds of uric acid is excreted by the kidneys and one-third by the small intestine. Uric acid is filtered through the glomeruli and reabsorbed in the proximal tubules. The excretion rate of uric acid in the proximal tubules of the kidney is approximately 10% [20] and is regulated by transporters expressed at the apical and basolateral sides. Uric acid reabsorption is mediated by URAT1/*SLC22A12*, OAT4/*SLC22A11*, and OAT10/*SLC22A3* transporters at apical sides [21], and by GLUT9/*SLC2A9* transporter at basolateral sides. Uric acid secretion is mediated by OAT1/*SLC22A11* and OAT3/*SLC22A8* transporters at the basolateral sides and by NPT1/*SLC17A1*, NPT4/*SLC17A3*, MRP4/*ABCC4*, and BCRP/*ABCG2* at the apical sides [22,23] (Figure 1). URAT1 and GLUT9 are transporters for uric acid reabsorption, and their dysfunction causes hypouricemia. Patients with URAT1 and GLUT9 dysfunctional variants are called hereditary renal hypouricemia types 1 and 2, respectively. In the intestine, BCRP/*ABCG2* secretes uric acid into the intestinal lumen [24]. Genome-wide association studies have reported over 30 genetic variants in uric acid transporters that affect serum uric acid. Three uric acid transporters, URAT1, GLUT9, and ABCG2, play key roles in the regulation of serum uric acid, and their dysfunctions lead to dysuricemia (hypouricemia and hyperuricemia). The role of *ABCG2* variants has been shown to be more important for the risk of hyperuricemia than environmental factors such as obesity and intense alcohol consumption [25]. Nonsynonymous allelic variants of *ABCG2* were shown to significantly hasten the onset of hyperuricemia and increase the likelihood of gout and the presence of a family history of gout. *ABCG2* dysfunction was known to be a risk factor for pediatric-onset hyperuricemia and gout [26]. Furthermore, Nakayama et al. reported a significant increase in the selection of the *ABCG2* and *ALDH2* loci in gout patients in Japan [27].

Serum uric acid levels are affected by several drugs through transporters. Loop diuretics and thiazide diuretics decrease uric acid excretion by both decreasing extracellular fluid volume and glomerular filtration rate and inhibiting uric acid secretion via NPT4 [28,29]. Losartan, an angiotensin II receptor blocker (ARB) [23,30], and calcium channel blockers (CCBs) [31] also inhibit URAT1 and increase uric acid excretion. Cyclosporin A promotes uric acid uptake via OAT10 [32]. Sodium/glucose co-transporter-2 (SGLT2) inhibitors are agents that increase urinary glucose. Since glucose and uric acid are co-transported with glucose, serum uric acid levels decreased [14,33]. Sacubitril/valsartan also reduces serum uric acid [34]. However, the mechanism by which they lower uric acid remains unclear.

Uric acid is a weak acid with a pKa of 5.8. Uric acid occurs primarily as an anionic urate at physiological pH of 7.4. The reference range for serum uric acid in humans is 1.5–6.0 mg/dL for women and 2.5–7.0 mg/dL for men. There is a difference between the sexes in that hyperuricemia is more common in men. Uric acid has low solubility in water, so the average concentration of uric acid in human serum is at the dissolution limit (6.8 mg/dL). When this level is exceeded, it is crystallized as monosodium urate (MSU) [17]. In most species, uric acid is broken down into allantoin by urate oxidase (uricase). However, humans have lost uricase, so they exhibit higher levels of uric acid than other mammals [35].

#### 2.1.1. The Pros and Cons of Uric Acid: The Pros

Over the course of human evolution, mutations in genes involved in ascorbic acid synthesis caused loss of function. Simultaneously, mutations in the uricase gene led to uric acid becoming the final metabolite in this pathway. Ames et al. showed that uric acid might act as an antioxidant in various redox reactions [36]. Uric and ascorbic acids are considered the most important water-soluble antioxidants [37], and the plasma level of uric acid is approximately six times that of ascorbic acid [38]. Ascorbic acid has a tendency to undergo oxidization and mutation, suggesting that the resulting mutated metabolite may contain peroxide radicals. Therefore, uric acid is considered a better antioxidant than ascorbic acid [39]. In addition, Yeum et al. measured the antioxidant capacity of the human body and suggested that uric acid accounts for 60% of the total antioxidant capacity in plasma [40]. This suggests that uric acid may provide the major endogenous defense against oxidative damage in the body [41]. Fabbrini showed that uric acid is a major antioxidant that may protect against oxidative damage caused by free radicals [42]. It is widely accepted that high levels of blood uric acid are an evolutionary advantage in humans because uric acid protects the heart, blood vessels, and nerve cells from oxidative damage. Uric acid also acts as a defense against aging and cancer by preventing oxidative damage caused by ROS [43] (Table 1) (Figure 2).

Uric acid reacts with a variety of oxidants, among which it has been reported that it preferentially reacts with peroxynitrite to form triuret [44]. Peroxynitrite is thought to contribute to the development of cardiovascular disease and has been shown to cause oxidative damage via tissue nitration [45]. Uric acid has been suggested to act as a scavenger of peroxynitrite [46]. In addition, uric acid can help keep the levels of superoxide dismutase, an enzyme playing a key role in protecting the extracellular environment from oxidative stress. In the extracellular environment, uric acid can scavenge hydroxyl radicals and peroxynitrite, but it has been shown to lose antioxidant capacity in a hydrophobic environment [47,48]. A prospective case-control study by Nieto et al. showed an association between elevated total serum antioxidant capacity and elevated serum uric acid levels in patients with atherosclerosis and concluded that this elevated uric acid level is a compensatory mechanism to counteract the oxidative damage associated with atherosclerosis in humans [41].

It is well known that uric acid levels increase under starvation conditions [49]. First, fasting causes rapid weight loss, followed by a period of lipid utilization and a subsequent period of protein degradation. Serum uric acid levels can be markedly elevated during the proteolytic phase with weight loss, and these elevated uric acid levels are associated with increased exercise, decreased water excretion, and elevated cortisol levels [49]. Acute increases in uric acid levels induce locomotor activity, exploratory activity, and impulsivity in rats [50]. When food is scarce and sodium intake is low, uric acid may promote sodium retention and blood pressure maintenance, thus favoring survival [51]. Studies have shown that the increase in locomotor activity associated with feeding is mediated by an increase in corticosterone levels, which is associated with an increase in uric acid levels [52]. Because uric acid is chemically similar to caffeine, it has potential neurostimulatory effects, increasing human cognition, alertness, and motivation. Increased uric acid levels may be advantageous for survival [53]. In fact, we found a weak association among uric acid levels, IQ test results, and school performance [54,55].

Shi et al. reported that uric acid is an endogenous danger signal released from damaged cells. Damaged cells release uric acid, which stimulates dendritic cell maturation and promotes CD8^+^ T-cell responses [56]. In addition, Shi et al. identified uric acid as one of the endogenous adjuvants and revealed that the removal of uric acid reduced the production of cytotoxic T lymphocytes against antigens in the transplanted cells [57]. Thus, uric acid is involved in T-cell activation in various contexts, such as tumors, transplants, and autoimmunity.

#### 2.1.2. The Pros and Cons of Uric Acid: The Cons

Uric acid produces allantoin and peroxynitrite via superoxide radicals to form triuret. The chemical reaction of uric acid with peroxynitrite produces aminocarbonyl and triurethanol radicals, which react with myeloperoxidase to produce the pro-oxidant uric acid hydroperoxide [58]. In contrast, uric acid in cells stimulates reduced NADPH oxidase and functions as a pro-oxidant [46] (Table 1) (Figure 3). The effects of exogenous uric acid on cells can be prevented by inhibiting the uptake of uric acid into the cells. Similarly, the biological effects of endogenous uric acid can be prevented by inhibiting its synthesis using XOR inhibitors [59]. Uric acid exhibits autocrine, paracrine, and endocrine activities. Intracellular uric acid stimulates proinflammatory transcription factors, growth factors, vasoconstrictive substances (angiotensin II, thromboxane, and endothelin), and chemokines, leading to mitochondrial dysfunction [58,59]. Uric acid also reduces endothelial NO bioavailability through various mechanisms, causing endothelial cell damage [60,61]. Thus, depending on the chemical environment, uric acid may not be an antioxidant but a pro-oxidant.

Furthermore, uric acid induces the growth of vascular smooth muscle cells (VSMCs) and triggers activation of the extracellular signal-regulated kinase (ERK), mitogen-activated protein (MAP) kinases, and platelet-derived growth factor (PDGF). Uric acid also induces the inflammatory pathway in VSMCs and increases the production of monocyte chemoattractant protein-1 (MCP-1). It has also been shown to cause the activation of cyclooxygenase-2 (COX-2), p38 MAPK, nuclear transcription factor nuclear factor-kappa B (NF-κB), and activator protein-1 (AP-1) [58]. According to a previous study, soluble uric acid, as well as uric acid crystals, can cause inflammation. Corry et al. also showed that uric acid stimulates VSMC proliferation, angiotensin II production, and oxidative stress via the vascular renin-angiotensin system (RAS) in tissues [37]. They also showed that uric acid causes cardiovascular damage by stimulating vascular RAS through the MAP kinase pathway [37]. Sautin et al. reported that uric acid taken up by adipocytes increases intracellular reactive oxygen species (ROS) production in differentiated adipocytes through the activation of nicotinamide adenine dinucleotide phosphate (NADPH) oxidase [47].

### 2.2. Hypertension

Numerous epidemiological studies have reported that elevated uric acid is an independent predictor of hypertension [62]. The relationship between serum uric acid levels and blood pressure was found to be dose-dependent and linear. Hyperuricemia is a strong independent predictor of hypertension with an approximately 2-fold increased risk within 5–10 years [63,64,65]. Hyperuricemia induced by uricase inhibitors in rats leads to hypertension. The mechanism by which hyperuricemia causes hypertension involves the sympathetic nervous system hyperactivity, endothelial dysfunction with reduced NO levels, and renal vasoconstriction mediated by activation of the renin-angiotensin system [66]. Rats with hyperuricemia-induced hypertension were reported to develop hypertension on a high-salt diet even after the uricase inhibitor was discontinued [51]. The mechanism behind this salt sensitivity was considered to involve uric acid entering vascular smooth muscle cells, triggering cell proliferation. In addition, hyperuricemia is thought to activate the renin-angiotensin system and produce inflammatory mediators. As a result, hyperuricemia leads to renal microangiopathy and narrowing of the arterial lumen [51]. Uric acid-lowering drugs can initially ameliorate hypertension; however, as kidney disease progresses, hypertension becomes salt-sensitive and uric acid-independent [67]. In other words, the pathogenesis of hypertension is thought to begin with the development of renal vasoconstriction, ischemia, and oxidative stress by uric acid, followed by persistent inflammation and organic changes (especially tubulointerstitial changes), resulting in increased blood pressure [68].

In a study of new-onset essential hypertension by Feig et al., 90% of subjects with essential hypertension and 30% of those with secondary hypertension had elevated serum uric acid levels (>5.5 mg/dL) compared with 0% of controls, revealing a clear association between uric acid and hypertension [69]. Kuwabara et al. also found that, for every 1 mg/dL decrease in serum uric acid in healthy male subjects, the protective effect against hypertension increased by 18%. In healthy female subjects, each 1 mg/dL decrease in serum uric acid levels was associated with a 31% decrease in the development of hypertension [70]. Elsewhere, the effect of hypouricemic therapy on blood pressure was observed in a randomized, double-blind, placebo-controlled trial. Sixty prehypertensive obese adolescents received allopurinol, probenecid, or a placebo, respectively. After 2 months, systolic blood pressure had increased from baseline by +1.9 mmHg (95% CI: −0.4 to 2.4 mmHg) in the placebo group, while it had decreased by −9.2 mmHg (−6.7 to −11.3 mmHg) in the allopurinol group and by −8.9 mmHg (−7.0 to −10.9 mmHg) in the probenecid group [71]. In addition, in a randomized, double-blind, placebo-controlled crossover trial conducted by Feig et al., allopurinol normalized blood pressure in 19 of 22 subjects who achieved a serum uric acid level of <5.0 mg/dL, whereas only 1 of 30 subjects achieved normal blood pressure with placebo treatment. This indicates that lowering uric acid levels with allopurinol lowers blood pressure in patients with hyperuricemia [72]. In this study, patients with hyperuricemia and hypertension tended to have elevated plasma renin activity, and xanthine oxidase inhibitors reduced plasma renin activity and plasma aldosterone [72,73].

Meanwhile, the PIUMA study showed a J-shaped relationship between uric acid levels and the incidence of cardiovascular disease events in both men and women [74]. The association between serum uric acid levels and cardiovascular events was also demonstrated in the Syst-Eur study in elderly subjects with systolic hypertension, in which a J-shaped curve was observed between serum uric acid levels and all-cause mortality [75]. Furthermore, in a cardiovascular study in the elderly (CASTEL), serum uric acid levels independently predicted coronary mortality in a J-shaped manner in the elderly with diabetes [76]. Kawasoe et al. used large-scale health checkup data to examine the association between serum uric acid levels and the prevalence of blood pressure abnormalities, considering the effects of other risk factors. The prevalence of hypertension was lowest in the 2.1–4.0 mg/dL serum uric acid group (second-lowest serum uric acid levels). However, the incidence of hypertension was significantly higher in the 4.1–5.0, 5.1–6.0, and ≥6.1 mg/dL serum uric acid groups, as well as the ≤2.0 mg/dL serum uric acid group (lowest serum uric acid levels). High and low serum uric acid levels were significantly associated with an increased prevalence of hypertension, creating a J-shaped curve [77]. Uric acid has antioxidant effects and contributes significantly to ROS removal. One of the etiologies of hypertension is oxidative stress, which results in the excessive generation of ROS. Elevated ROS levels contribute to vascular dysfunction and remodeling because of oxidative damage [78]. However, there is no clear evidence that oxidative stress alone contributes to the pathogenesis of hypertension; in addition to increased oxidative stress due to hypouricemia, it is probably associated with elevated blood pressure, including other factors of hypertension (salt, the renin-angiotensin system, and symptomatic hyperactivity) [79]. Increased oxidative stress due to decreased uric acid levels may inactivate the nitric oxide released by the endothelium and inhibit vasorelaxation. Therefore, the relationship between blood pressure and serum uric acid levels might have developed a J-shaped curve [77].

### 2.3. Diabetes Mellitus and Metabolic Syndrome

Hyperuricemia is closely associated with metabolic abnormalities, such as obesity, dyslipidemia, insulin resistance, type 2 diabetes, and metabolic syndrome [80]. Elevated serum uric acid levels are surrogate markers of metabolic syndrome [81]. Japanese researchers surveyed a group of healthy young males for 5 years and revealed that those with high serum uric acid levels gained four times as much weight as those with low levels [6]. Therefore, elevated serum uric acid levels may trigger obesity and metabolic syndrome. Furthermore, increased serum uric acid concentration has been shown to be associated with an increased risk of insulin resistance [82].

Fructose intake induces fatty liver, elevated triglycerides, and elevated serum uric acid [83,84]. Once inside the cell, fructose is rapidly phosphorylated by fructokinase. During fructose metabolism, cellular energy (ATP) is reduced; however, ATP breakdown products (such as AMP) enter the nucleotide degradation pathway to produce uric acid. Serum uric acid levels increase within minutes of fructose ingestion and are often associated with lactate release [85]. Fructose stimulates appetite and induces insulin resistance to raise glucose levels in the body, thereby providing glucose to the brain. Similarly, it stimulates water intake and increases blood pressure to reduce sodium loss [86].

Choi et al. reported a large prospective cohort study of male patients at high cardiovascular risk. The results showed that men with gout had a high risk of developing type 2 diabetes in the future, independent of other risk factors [87]. The Atherosclerosis Risk in Communities Study in the United States, which followed nondiabetic patients with hyperinsulinemia for 11 years, found that hyperuricemia was an independent risk factor for progression to hyperinsulinemia [88].

Another study suggested that the kidney may be involved in the association between hyperuricemia and the risk of type 2 diabetes mellitus [89]. Renal tubular function is affected by hyperinsulinemia, and decreased insulin-mediated glucose metabolism results in decreased uric acid clearance. Thus, reduced uric acid excretion causes hyperuricemia [89,90]. Insulin may promote renal uric acid reabsorption via URAT1 or Na^+^-dependent anion co-transporter in the proximal renal tubules [91]. In addition, insulin is responsible for inducing hyperuricemia in the renal tubules [92]. Conversely, a decrease in serum uric acid levels has been observed, and there is a bell-shaped relationship between uric acid levels and hemoglobin A1c. This is probably due to the fact that the rate of renal uric acid excretion depends on the level of glycemic control [93]. In addition, Lytvyn et al. reported that glycosuria was induced by the inhibition of sodium/glucose co-transporter 2 (SGLT2) in patients with type 1 diabetes, resulting in a decrease in blood uric acid levels and an increase in the rate of uric acid excretion [94]. Patients whose uric acid levels increase as glycemic control improves are often encountered. This is thought to be due to decreased reabsorption of uric acid in the proximal tubules as insulin levels decrease. Meanwhile, urinary excretion of uric acid has been shown to increase during hyperglycemia, and the mechanism behind this is assumed to involve osmotic diuresis [14]. SGLT2 inhibitors lower blood uric acid levels by increasing urinary glucose and uric acid excretion co-transported with glucose [95,96]. In addition, patients with familial renal diabetes and SGLT2 mutations tend to have lower serum uric acid levels [33]. Chino et al. reported that the SGLT2 inhibitor luseogliflozin administered to healthy subjects caused a decrease in serum uric acid levels and an increase in the excretion rate of urinary uric acid. However, no inhibition of uptake of the major renal tubular uric acid reabsorption transporter was observed [97]. Thus, whether uric acid excretion is due to osmotic pressure or to transporters such as SGLT2 remains unknown.

### 2.4. Neurological Disease and Mental Illness

As hyperuricemia is associated with the development of various cardiovascular diseases, it is possible that hyperuricemia and cardiovascular diseases are involved in stroke-related deaths. The Reasons for Geographic and Racial Differences in Stroke (REGARDS) study, conducted in 2020, was a case-cohort study performed using a large dataset. The study concluded that hyperuricemia might be a risk factor for stroke [98]. Furthermore, several epidemiological studies have shown a J-shaped association between serum uric acid levels and stroke events [74,99]. A longitudinal national cohort study by Kamei et al. reported that elevated serum uric acid levels of 7.1 mg/dL or higher in men and 5.5 mg/dL or higher in women were independently associated with the incidence of nonfatal stroke. Similarly, in the present study, even the group with low uric acid levels had an increased risk of stroke, showing a J-shaped curve [99].

Reactive oxygen plays a major role in the development of neurodegenerative disease because a large number of reactive oxygen species are produced in the brain. Despite accounting for only 2% of the body’s weight, the brain consumes 20% of the body’s oxygen supply, and most of the oxygen is converted to ROS [100]. The concentration of uric acid in the cerebrospinal fluid is always low, approximately ten times that in the peripheral blood. This concentration in the CSF depends on the blood uric acid level and the integrity of the blood-CSF barrier [37].

Hyperuricemia has been shown to have neuroprotective effects in neurodegenerative diseases, such as Parkinson’s disease and multiple sclerosis, and in dementia, such as Alzheimer’s disease. Elevated blood uric acid levels have been shown to be associated with a reduced risk of developing Parkinson’s disease [101]. Patients with multiple sclerosis and neuromyelitis optica spectrum disorders have lower serum uric acid levels than healthy controls, indicating that serum uric acid levels are a potential diagnostic biomarker [102]. It was also shown that the prevalence of amyotrophic lateral sclerosis in patients with gout was significantly lower than in patients without gout [103]. We also found a reduced risk of dementia in patients with high serum uric acid levels in dementia [104]. There is also an inverse relationship between gout and the risk of developing Alzheimer’s disease, supporting the possible neuroprotective role of uric acid [105].

Patients with Alzheimer’s disease- and Parkinson’s disease-related dementia showed a strong association with high serum uric acid levels; however, no correlation was found in patients with vascular or mixed dementia, suggesting a different relationship with serum uric acid levels, especially in patients with vascular dementia [106]. This suggests that although the antioxidant effect of uric acid prevents nerve degeneration and the development of dementia, both effects need to be considered because hyperuricemia promotes atherosclerosis and adversely affects cognitive function.

Serum uric acid levels are elevated in subjects with anger management issues, while those who exhibit aggressive behavior, including criminal behavior, have higher serum uric acid levels [107]. High urinary uric acid levels have also been associated with aggressive behavior [108]. Moreover, those with Lesch–Nyhan disease, an inherited form of hyperuricemia, show impulsivity, aggression, and a risk of self-harm [109]. It was reported that uric acid-lowering therapy with allopurinol effectively reduces impulsive or aggressive behavior and self-harm in some patients [110]. Conversely, some have suggested that abnormal function of the dopamine pathway in the basal ganglia is a cause of self-injurious behavior, not uric acid [111]. In addition, patients with bipolar mania often have elevated serum uric acid due to dysfunction of the purinergic system, with a decrease in adenosine neurotransmission [112]. Conversely, depression is associated with a decrease in serum uric acid levels, which may represent a marker of the condition [113]. Although there are reports of patients with primary hypouricemia, such as type 1 xanthinuria, an XOR deficiency [114], and renal hypouricemia, a disorder of URAT1 and GLUT9 [115,116], no reports have been published of neurological symptoms in these patients. This suggests that uric acid is not cerebroprotective and may support the value of serum uric acid levels as a diagnostic biomarker in cranial nerve disease.

### 2.5. Chronic Kidney Disease (CKD)

Elevated serum uric acid levels are widely accepted to predict the development of CKD [117], but whether they are a risk factor or a marker of CKD is controversial. Messerli et al. reported that renal blood flow decreased, and renal vascular resistance and total peripheral resistance increased in patients with hyperuricemia [118]. Elevated serum uric acid is associated with renal lesions such as glomerulosclerosis and interstitial fibrosis because of the deposition of uric acid crystals, suggesting that hyperuricemia in hypertensive patients may influence nephrosclerosis [118]. Serum uric acid was also suggested to be involved in CKD in a large cohort study [119]. In addition, a prospective observational cohort of normotensive healthy individuals with normal renal function showed an association between elevated serum uric acid levels and renal impairment [120]. A meta-analysis by Luo et al. also reported that elevated serum uric acid levels were associated with an increased risk of cardiovascular mortality in patients with CKD [121]. Moreover, Kuwabara et al. reported that, among healthy subjects, each 1 mg/dL decrease in serum uric acid in men was associated with a 23% increase in the protective effect against CKD, and each 1 mg/dL decrease in serum uric acid in women was associated with a 42% increase in the prevention of CKD [70].

Meanwhile, a 10-year follow-up study of routine health check-up participants by Mori et al. showed a U-shaped relationship between uric acid levels and the risk of CKD in women [122]. In that study, the risk of CKD in men increased with higher levels of uric acid but not with lower levels. Therefore, only in women was a low uric acid level a significant risk factor for CKD. A large cross-sectional study by Kuwabara et al. showed that patients with hypouricemia had a 9-fold higher incidence of kidney disease than patients with no hypouricemia [70]. That study also examined the cumulative incidence of cardiometabolic disease over a 5-year period in patients with hypouricemia and normouricemia (3–5 mg/dL in men and 2–4 mg/dL in women) and indicated that hypouricemia was a risk factor for the development of hypertension and CKD only in women [70]. Renal hypouricemia is a disease associated with the increased excretion of uric acid in the kidneys. Patients with renal hypouricemia can develop urolithiasis and exercise-induced acute kidney injury (EIAKI) [123]. EIAKI is considered a transient acute kidney injury that does not cause severe kidney dysfunction over an extended period.

Whether interventions with uric acid-lowering drugs would reduce renal dysfunction is unclear. In recent years, two clinical studies on patients with asymptomatic hyperuricemia without gout have been reported in Japan [124,125]. In the FEATHER Study, febuxostat, an XOR inhibitor, did not improve renal function compared with placebo in patients with CKD and hyperuricemia; however, subgroup analyses showed that it significantly reduced renal function decline in patients without proteinuria and in patients with serum creatinine levels below the median [124]. The FREED Study demonstrated that febuxostat significantly reduced brain, cardiovascular, and renal events compared with conventional therapy with lifestyle modifications in patients with asymptomatic hyperuricemia aged 65 years and older [125].

### 2.6. Cardiovascular Disease

Longitudinal data from the NHANES showed that all-cause mortality and cardiovascular mortality were associated with serum uric acid levels in both men and women at 10 years of follow-up [126]. Serum uric acid levels may be associated with cardiovascular disease and prognosis, but their role remains controversial. The Framingham Heart Study showed that serum uric acid levels were not causally related to the development of coronary heart disease, death from cardiovascular disease, or death from all causes [127]. Some studies have shown that serum uric acid levels are a strong predictor of mortality from cardiovascular diseases [128,129,130]. Furthermore, it has been reported that, in humans, treatment with high doses of allopurinol improves endothelial cell function not by lowering plasma uric acid levels but by lowering oxidative stress in the blood vessels [131].

Meanwhile, there is a J-shaped relationship between serum uric acid and cardiovascular disease, and attention has recently been paid to the idea that both low and high uric acid levels are associated with a greater risk of developing cardiovascular events [132]. Vascular endothelial dysfunction due to decreased antioxidant activity associated with uric acid levels has been suggested as a factor contributing to an increased risk of cardiovascular events caused by hypouricemia [133]. The PIUMA study showed that serum uric acid levels lower than 4.5 mg/dL in men and 3.2 mg/dL in women with essential hypertension increased the incidence of cardiovascular disease and its-related deaths [74]. That is, it was suggested that there is an inverse relationship between serum uric acid levels and the incidence of cardiovascular disease among subjects with serum uric acid levels below 4.5 mg/dL in men and 3.2 mg/dL in women [74,75,76].

### 2.7. COVID-19

Some previous studies showed associations between high [134,135] and low uric acid levels [136,137,138] with COVID-19 severity. For example, Fukushima et al. investigated the relationship between uric acid levels and COVID-19 severity in a Japanese cohort and reported that uric acid levels of 7.6 mg/dL or higher and uric acid levels of 2.5 mg or lower were associated with COVID-19 severity. This indicated a U-shaped relationship between both low and high serum uric acid levels and COVID-19 severity [139].

## 3. Conclusions

In addition to causing gout, hyperuricemia has been implicated in various cardiovascular and renal diseases, such as hypertension, arteriosclerosis, kidney disease, and cardiovascular disease. In the 1900s, it was believed that uric acid was not a true risk factor for developing cardiovascular or renal diseases [140]; consequently, uric acid levels were not assessed during routine blood tests. Additionally, routine assessment of uric acid levels was not recommended for patients with cardiovascular or kidney disease, and this level was eliminated as a risk factor from the lists provided by most cardiovascular and kidney societies. Therefore, uric acid was ultimately disregarded as a risk factor for kidney diseases [4]. However, in the late 1990s, uric acid was re-examined as a risk factor for cardiovascular and renal diseases, and in recent years, there has been growing interest in investigating the role of uric acid owing to the increased prevalence of hyperuricemia worldwide. The dual properties of uric acid create confusion regarding its effects on various diseases. Soluble uric acid is not an inert substance and has been found to possess a number of biological properties [141]. It has become clear that uric acid has two contradictory properties: on the one hand, it acts as an antioxidant, while on the other hand, it acts as an oxidant. Thus, although it was previously thought that high uric acid levels would have negative effects on hypertension and cardiovascular diseases, it is also important to consider the effects of low uric acid levels. Although many clinical trials have been conducted on uric acid, the causal relationship between uric acid and hypertension or cardiovascular disease remains unclear, probably because of its biological properties. Therefore, if we propose the concept of “dysuricemia,” which encompasses both hyperuricemia and hypouricemia and consider the involvement of this condition in various diseases, we believe that further research on uric acid will be enhanced. In conclusion, we suggest that future clinical studies focusing not only on high uric acid levels but also on low uric acid levels are needed.

## Figures and Tables

**Figure 1 biomedicines-11-01255-f001:**
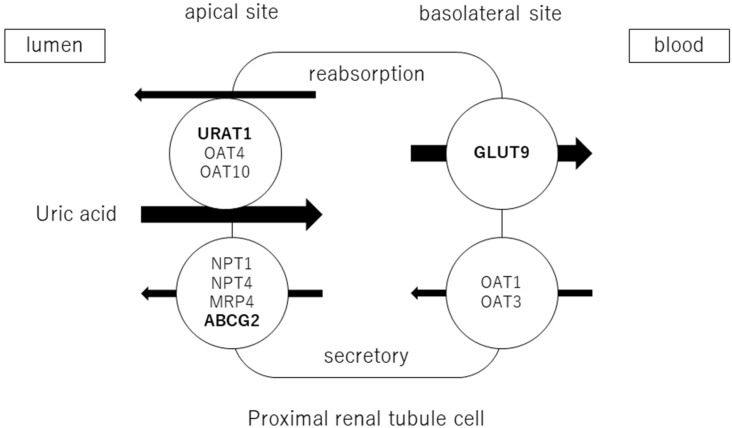
Uric acid transport molecules in the proximal renal tubules. Uric acid is regulated by several transporters expressed at the apical and basolateral sides. Uric acid reabsorption is mediated by URAT1/*SLC22A12*, OAT4/*SLC22A11*, and OAT10/*SLC22A3* transporters at the apical side, and GLUT9/*SLC2A9* transporter at the basolateral side. Uric acid secretion is mediated by OAT1/*SLC22A11* and OAT3/*SLC22A8* transporters at the basolateral side, and NPT1/*SLC17A1*, NPT4/*SLC17A3*, MRP4/*ABCC4*, and BCRP/*ABCG2* at the apical side. URAT1, GLUT9, and ABCG2 were bolded, as they play crucial roles in the regulation of serum uric acid, and their dysfunctions cause uric acid transport disorders (hypouricemia and hyperuricemia).

**Figure 2 biomedicines-11-01255-f002:**
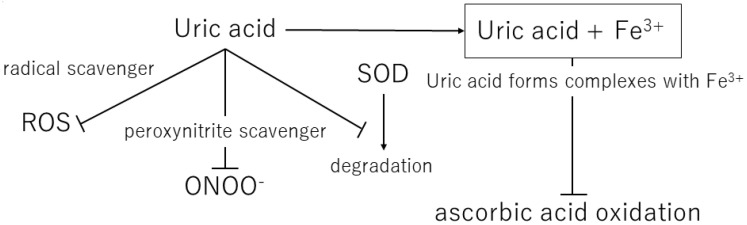
Mechanisms of the antioxidant effects of extracellular uric acid. ROS, reactive oxygen species; ONOO^−^, peroxynitrite; SOD, superoxide dismutase.

**Figure 3 biomedicines-11-01255-f003:**
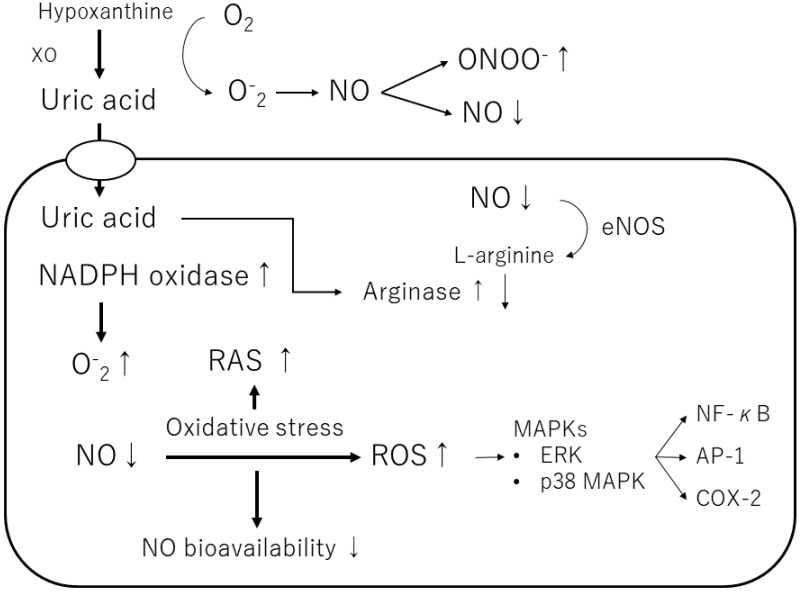
Mechanisms of the pro-oxidant effect of intracellular uric acid. Intracellular uric acid induces reactive oxygen species (ROS) production and activates several signaling pathways. XO, xanthine oxidase; NO, nitric oxide; ONOO^−^, peroxynitrite; NADPH oxidase, nicotinamide adenine dinucleotide phosphate oxidase; eNOS, endothelial NO synthase; ERK, extracellular signaling-regulated kinase; MAPK, mitogen-activated protein kinase; RAS, renin-angiotensin-aldosterone system; NF-κB, nuclear factor-kappa B; AP-1, activator protein 1; COX-2, cyclooxygenase 2.

**Table 1 biomedicines-11-01255-t001:** Mechanisms of uric acid: antioxidant effects and pro-oxidant effects.

Antioxidant effects of uric acid
1. Prevents oxidative damage caused by reactive oxygen species
2. Uric acid works as a scavenger of peroxynitrite
3. Inhibits degradation of superoxide dismutase (SOD)
4. Uric acid forms a complex with Fe^3+^ (works as achelator of Fe^3+^)
Pro-oxidant effects of uric acid
1. Uric acid is taken up into the cell via the transporters and produces ROS via activation of nicotinamide adenine dinucleotide phosphate (NAPDH) oxidase

## Data Availability

Data sharing not applicable. No new data were created or analyzed in this study. Data sharing is not applicable to this article.

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
