# Peer review of "Dysuricemia—A New Concept Encompassing Hyperuricemia and Hypouricemia"

_biomedicines, 2023, doi:10.3390/biomedicines11051255_

Round 1
Reviewer 1 Report
The authors present a review of dysuricemia. The topic is interesting, however, the manuscript would be strengthened considerably by the revision in response to the following comments:
Urate homeostasis depends on a balance between production and the complicated process of secretion and reabsorption in the renal proximal tubule and excretion in the intestines. Genome-wide association studies and meta-analyses have revealed over 30 common genetic variants influencing serum urate, mostly in urate transporters. Max. 10% of hyperuricemia exist to mendelian disorders of purine metabolism as overactivity of phosphoribosylpyrophosphate synthetase (OMIM #300661) and deficiency of hypoxanthine-guanine phosphoribosyltransferase (OMIM 308000).
Three urate transporters, URAT1/SLC22A12, GLUT9/SLC2A9, and ABCG2/BCRP have been reported to play crucial roles in the regulation of SU, and their dysfunctions cause urate transport disorders (hypouricemia and/or hyperuricemia). The heritable secretion component of urate homeostasis is principally mediated by the product of the ABCG2/BCRP gene. On the other hand, the dysfunctional variants in URAT1 and GLUT9 cause mendelian hereditary renal hypouricemia type 1 and 2.
It would be appropriate to emphasize the role of ABCG2 variants which have been shown to have stronger effects on the risk of hyperuricemia than major environmental risks factors such as obesity and heavy drinking (Sci Rep 2014;4:5227). The common polymorphism rs2231142, allelic variant p.Q141K, results in a 53% reduction in UA transport with at least 10% of all gout cases in people of European ancestry attributable to this variant. Non-synonymous allelic variants of ABCG2 had a significant effect on the earlier onset of hyperuricemia/gout and the presence of a familial gout history, ABCG2 dysfunction was reported as a strong independent risk for pediatric-onset hyperuricemia/gout (Arthritis Research & Therapy 2019; 21:77). Moreover, was reported selection pressure analysis revealed significant enrichment of selection for the ABCG2 and ALDH2 loci in Japanese gout patients (Ann Rheum Dis. 2020 May;79(5):657-665) - here may be a combination of both mechanisms related to hyperuricemia. ABCG2 dysfunction significantly elevated serum UA in acute gastroenteritis patients regardless of the degree of dehydration, which demonstrated the pathophysiological role of ABCG2 in acute gastroenteritis (importance of intestinal epithelium as an excretion pathway besides an absorption pathway)
It is also important to mention that SLC22A12 and SLC2A9 is referred to as genes that are associated not only with hyperuricemia and gout, but also with hypouricemia since it is urate reuptake transporter. At present, more than 50 variations in the SLC22A12 coding region have been described and most of the variants are currently associated with the hypouricemia phenotype. RHUC patients have been described in different ethnic groups and geographically noncontiguous countries with significant population specificity: The high incidence of RHUC1 has been reported in the Asia region and is attributed to the high frequency (2.30–2.37 %) of the c.774G>A (p.W258X) and (0.40 %) c.269G>A (p.R90H) in SLC22A12 gene among Japanese and Koreans, with null allele frequency in African/American, Ashkenazi Jewish, South Asia, and European population, which is indicative of a founder mutation on the Asian continent. However, the world's highest frequency of predominant dysfunctional URAT1 variants was identified in the European Roma population: the p.T467M variant (5.6%), and the deletion variant p.L415_G417 (1.9%).
Authors reported that hyperuricemia may cause side effects in humans with psychiatric disorders. However: primary hypouricemia is a characteristic of xantine dehydrogenase (XDH, E.C. 1.1.1.204) deficiency (OMIM #278300, OMIM 603592) and renal hypouricemia type 1 and 2 (RHUC). In these patients neurological symptoms were not observed – including cases with extremely low serum UA with values near 0. These findings suggest that the protective systems involving plasma uric acid are not essential. This discrepancy should be discussed.
Hypoxanthine-guanine phosphoribosyltransferase deficiency is one of the most common inborn errors of purine metabolism. This X-linked disorder (OMIM 308000) is classified into distinct forms. Partial deficiency (#300323) is associated with a clinical manifestation of purine overproduction that results in increased urate synthesis – patients are at risk of gout and urate nephrolithiasis/urolithiasis. However, some patients develop a variable spectrum of neurological manifestations, such as motor disability and intellectual impairment, (Lesch–Nyhan variants). Classical features of severe deficiency, Lesch–Nyhan syndrome (#300322), are characterized by neurological and behavioral abnormalities and the overproduction of urate. HGprt deficiency is associated with a selective reduction in dopamine levels in the brain’s basal ganglia - HGprt deficiency causes a change in integrity of the dopaminergic neurochemical phenotype. Thus, this mechanism is causal for neurological impairment in patients, not hyperuricemia.
Some of the information given would deserve a link to the relevant references - for example, „UA levels are elevated in psychiatric disorders characterized by high impulsivity..."
Use for the name of gene Italica font.
Author Response
Thank you for comments us to revise our manuscript entitled, “Dysuricemia – A new concept encompassing hyperuricemia and hypouricemia”. We also appreciate the time and effort you and each of the reviewers have dedicated to providing insightful feedback on ways to strengthen our paper. Thus, it is with great pleasure that we resubmit our article for further consideration. We have incorporated changes that reflect the detailed suggestions you have graciously provided. We also hope that our edits and the responses we provide below satisfactorily address all the issues and concerns you have noted.
Reviewer 1
The authors present a review of dysuricemia. The topic is interesting, however, the manuscript would be strengthened considerably by the revision in response to the following comments:
Comments:
Urate homeostasis depends on a balance between production and the complicated process of secretion and reabsorption in the renal proximal tubule and excretion in the intestines. Genome-wide association studies and meta-analyses have revealed over 30 common genetic variants influencing serum urate, mostly in urate transporters. Max. 10% of hyperuricemia exist to mendelian disorders of purine metabolism as overactivity of phosphoribosylpyrophosphate synthetase (OMIM #300661) and deficiency of hypoxanthine-guanine phosphoribosyltransferase (OMIM 308000).
Three urate transporters, URAT1/SLC22A12, GLUT9/SLC2A9, and ABCG2/BCRP have been reported to play crucial roles in the regulation of SU, and their dysfunctions cause urate transport disorders (hypouricemia and/or hyperuricemia). The heritable secretion component of urate homeostasis is principally mediated by the product of the ABCG2/BCRP gene. On the other hand, the dysfunctional variants in URAT1 and GLUT9 cause mendelian hereditary renal hypouricemia type 1 and 2.
It would be appropriate to emphasize the role of ABCG2 variants which have been shown to have stronger effects on the risk of hyperuricemia than major environmental risks factors such as obesity and heavy drinking (Sci Rep 2014;4:5227). The common polymorphism rs2231142, allelic variant p.Q141K, results in a 53% reduction in UA transport with at least 10% of all gout cases in people of European ancestry attributable to this variant. Non-synonymous allelic variants of ABCG2 had a significant effect on the earlier onset of hyperuricemia/gout and the presence of a familial gout history, ABCG2 dysfunction was reported as a strong independent risk for pediatric-onset hyperuricemia/gout (Arthritis Research & Therapy 2019; 21:77). Moreover, was reported selection pressure analysis revealed significant enrichment of selection for the ABCG2 and ALDH2 loci in Japanese gout patients (Ann Rheum Dis. 2020 May;79(5):657-665) - here may be a combination of both mechanisms related to hyperuricemia. ABCG2 dysfunction significantly elevated serum UA in acute gastroenteritis patients regardless of the degree of dehydration, which demonstrated the pathophysiological role of ABCG2 in acute gastroenteritis (importance of intestinal epithelium as an excretion pathway besides an absorption pathway)
It is also important to mention that SLC22A12 and SLC2A9 is referred to as genes that are associated not only with hyperuricemia and gout, but also with hypouricemia since it is urate reuptake transporter. At present, more than 50 variations in the SLC22A12 coding region have been described and most of the variants are currently associated with the hypouricemia phenotype. RHUC patients have been described in different ethnic groups and geographically noncontiguous countries with significant population specificity: The high incidence of RHUC1 has been reported in the Asia region and is attributed to the high frequency (2.30–2.37 %) of the c.774G>A (p.W258X) and (0.40 %) c.269G>A (p.R90H) in SLC22A12 gene among Japanese and Koreans, with null allele frequency in African/American, Ashkenazi Jewish, South Asia, and European population, which is indicative of a founder mutation on the Asian continent. However, the world's highest frequency of predominant dysfunctional URAT1 variants was identified in the European Roma population: the p.T467M variant (5.6%), and the deletion variant p.L415_G417 (1.9%).
Response:
Thank you for your informative comments. In accordance with the reviewer's comments, we have added the important urate transporters URAT1, GLUT9, and ABCG2 and made significant revisions to the manuscript to address this point (page 3, lines 18 to 56). In particular, URAT1, GLUT9, and ABCG2 were bolded for emphasis.
Comments:
Authors reported that hyperuricemia may cause side effects in humans with psychiatric disorders. However: primary hypouricemia is a characteristic of xantine dehydrogenase (XDH, E.C. 1.1.1.204) deficiency (OMIM #278300, OMIM 603592) and renal hypouricemia type 1 and 2 (RHUC). In these patients neurological symptoms were not observed – including cases with extremely low serum UA with values near 0. These findings suggest that the protective systems involving plasma uric acid are not essential. This discrepancy should be discussed.
Response:
In renal hypouricemia, neurologic symptoms are not observed despite low serum uric acid levels. These suggest that a protective system of uric acid is not essential. We fully agree with this reviewer's opinion. We have also discussed this discrepancy and incorporated it into the manuscript (page 8, line 48 to52).
Comments:
Hypoxanthine-guanine phosphoribosyltransferase deficiency is one of the most common inborn errors of purine metabolism. This X-linked disorder (OMIM 308000) is classified into distinct forms. Partial deficiency (#300323) is associated with a clinical manifestation of purine overproduction that results in increased urate synthesis – patients are at risk of gout and urate nephrolithiasis/urolithiasis. However, some patients develop a variable spectrum of neurological manifestations, such as motor disability and intellectual impairment, (Lesch–Nyhan variants). Classical features of severe deficiency, Lesch–Nyhan syndrome (#300322), are characterized by neurological and behavioral abnormalities and the overproduction of urate. HGprt deficiency is associated with a selective reduction in dopamine levels in the brain’s basal ganglia - HGprt deficiency causes a change in integrity of the dopaminergic neurochemical phenotype. Thus, this mechanism is causal for neurological impairment in patients, not hyperuricemia.
Some of the information given would deserve a link to the relevant references - for example, „UA levels are elevated in psychiatric disorders characterized by high impulsivity..."
Use for the name of gene Italica font.
Response:
The reviewers were correct and we have revised our manuscript.As noted by the reviewer, no psychiatric disorders are found in patients with primary hypouricemia (except for xanthinuria type 3). Therefore, in this manuscript, we treat uric acid as a biomarker for diagnosis, not as a mental effect of uric acid itself; we have modified the psychiatric symptoms of Lesch-Nyhan syndrome to be due to dopamine pathways (page 8, line 49 to page 9, line 18).
In addition, changed the name to gene Italic font.
Again, thank you for giving us the opportunity to strengthen our manuscript with your valuable comments and queries. We have worked hard to incorporate your feedback and hope that these revisions persuade you to accept our submission.

Reviewer 2 Report
In this review, the authors introduced the concept of "dysuricemia", a condition where a deviation from the appropriate range of uric acid results in disease. It's an interesting concept and is logically and coherently presented.
Have the authors considered all possible aspects? Here I have doubts.
Risk factors for hyperuricemia are more complex.
We can include here:
- so-called unmodifiable: male gender, old age, genetic predisposition
- Diseases and metabolic disorders: diabetes, obesity, overweight
Lipid metabolism disorders: hypercholesterolemia and hypertriglyceridemia
- Diet-related factors:
Purine-rich foods: meat, offal, seafood, substances that increase the catabolism of nucleotides:
fructose, alcohol (especially beer)
-Medicines: diuretics: loop diuretics and thiazide, beta blockers: propranolol, atenolol, metoprolol, cyclosporine, ethambutol, pyrazinamide, low doses of acetylsalicylic acid
Other crucial diseases/conditions not considered by the investigators, such as myelo- and lymphoproliferative diseases, polycythemia vera,
malignancy, hemolytic anemia, status after organ transplantation, radiotherapy, chemotherapy
Transient reasons, like significant physical exertion, dehydration, surgical procedure, injury, inflammation
It seems to me that there are many more conditions/disorders in this situation that should be addressed.
I couldn't find in the article a pathway to malignancy; what about medication and diet?
Besides, it seems that there is a lack of collective tables and figures so that the reader can sort everything out in his head. Hence, I suggest adding the omitted states at the end or simply addressing them in some other form, I leave it to the authors and suggest enriching the article with figures with metabolic pathways showing the essence of the presented concept.
Author Response
Thank you for comments us to revise our manuscript entitled, “Dysuricemia – A new concept encompassing hyperuricemia and hypouricemia”. We also appreciate the time and effort you and each of the reviewers have dedicated to providing insightful feedback on ways to strengthen our paper. Thus, it is with great pleasure that we resubmit our article for further consideration. We have incorporated changes that reflect the detailed suggestions you have graciously provided. We also hope that our edits and the responses we provide below satisfactorily address all the issues and concerns you have noted.
Reviewer 2
Comments:
In this review, the authors introduced the concept of "dysuricemia", a condition where a deviation from the appropriate range of uric acid results in disease. It's an interesting concept and is logically and coherently presented.
Have the authors considered all possible aspects? Here I have doubts.
Risk factors for hyperuricemia are more complex.
We can include here:
- so-called unmodifiable: male gender, old age, genetic predisposition
- Diseases and metabolic disorders: diabetes, obesity, overweight
Lipid metabolism disorders: hypercholesterolemia and hypertriglyceridemia
- Diet-related factors:
Purine-rich foods: meat, offal, seafood, substances that increase the catabolism of nucleotides:
fructose, alcohol (especially beer)
-Medicines: diuretics: loop diuretics and thiazide, beta blockers: propranolol, atenolol, metoprolol, cyclosporine, ethambutol, pyrazinamide, low doses of acetylsalicylic acid
Other crucial diseases/conditions not considered by the investigators, such as myelo- and lymphoproliferative diseases, polycythemia vera,
malignancy, hemolytic anemia, status after organ transplantation, radiotherapy, chemotherapy
Transient reasons, like significant physical exertion, dehydration, surgical procedure, injury, inflammation
It seems to me that there are many more conditions/disorders in this situation that should be addressed.
I couldn't find in the article a pathway to malignancy; what about medication and diet?
Besides, it seems that there is a lack of collective tables and figures so that the reader can sort everything out in his head. Hence, I suggest adding the omitted states at the end or simply addressing them in some other form, I leave it to the authors and suggest enriching the article with figures with metabolic pathways showing the essence of the presented concept.
Response:
The reviewers' opinions are helpful and appreciated. The reviewer advised us that risk factors for hyperuricemia are complex. We added additional information on gender differences in hyperuricemia (more common in men) (page 4, line 5), genetic predisposition (especially ABCG2) (page 3), dietary factors (page 3. line 4), and drug effect (page 4, line 1). In addition, we described about blood diseases (page 3, line 39 to 49). Our manuscript describes the effects of high or low uric acid on the human body (especially about disease). Therefore, we have not much to discuss the factors that could elevate uric acid (transient reasons, like significant physical exertion, dehydration, surgical procedure, injury, inflammation). Following the reviewer's advice, I added a table summarizing the antioxidant and oxidative effects of uric acid and a diagram of uric acid transporters to help the reader keep everything straight.
Again, thank you for giving us the opportunity to strengthen our manuscript with your valuable comments and queries. We have worked hard to incorporate your feedback and hope that these revisions persuade you to accept our submission.
